# Impacts of an Exercise Intervention on the Health of Pancreatic Beta-Cells: A Review

**DOI:** 10.3390/ijerph19127229

**Published:** 2022-06-13

**Authors:** Shuang Zhang, Yaru Wei, Chunxiao Wang

**Affiliations:** 1Department of Sports Science, University of Harbin Sport, Harbin 150008, China; zhangshuang1194@126.com (S.Z.); wyrwei2008@126.com (Y.W.); 2Department of Kinesiology, University of Shanghai Sport, Shanghai 200238, China

**Keywords:** exercise intervention, pancreas health, human, mice, rats, global health

## Abstract

There is an increasing consensus that exercise is a medicine and that regular exercise can effectively improve and prevent metabolic diseases such as diabetes. Islet cells are the endocrine of the pancreas and vital to the development of diabetes. Decades of developmental research in exercise intervention and the health of islet cells confirmed that exercise exerts beneficial effects on the function, proliferation, and survival rate of islet cells. However, the precise exercise reference scheme is still elusive. To accomplish this goal, we searched and analyzed relevant articles, and concluded the precise exercise prescription treatments for various species such as humans, rats, and mice. Each exercise protocol is shown in the tables below. These exercise protocols form a rich pipeline of therapeutic development for exercise on the health of islet cells.

## 1. Introduction

Diabetes mellitus is one of the major chronic non-communicable diseases in the world, reducing the quality of life. In addition, the high level of blood glucose and multiple complications also place substantial burdens on society, family, and the hospital system [1]. Rapid economic development and social burdens contribute to the explosive prevalence and mortality of diabetes mellitus globally [2]. 

According to the 2019 International Diabetes Federation statistics, more than 1 in 10 adults aged 20–70 years died from diabetes, which equates to eight deaths per minute. Moreover, diabetes prevalence has increased by 88% from 2006 to 2019. The increasing epidemic of diabetes mellitus has posed a serious health threat to the human population [3]. 

The pancreatic beta-cells are a key factor in the development of diabetes because of their capability of secreting insulin which lowers the blood glucose level [4]. Evidence confirmed that beta-cell loss could lead to type I diabetes mellitus (T1DM), and dysfunction of beta-cells leads to type II diabetes mellitus (T2DM) due to the lack of secreting enough insulin to reduce glucose levels. Hence, ensuring the integrity of beta-cells function is of great significance for the prevention and treatment of diabetes [4].

Exercise can effectively prevent or delay the incidence of type 2 diabetes [5]. In addition, regular physical activity also provides considerable benefits for cardiovascular health, muscle strength, and insulin sensitivity in type 1 diabetes mellitus patients [6]. Aerobic exercise combined with resistance training is the most effective exercise form to regulate blood glucose in adults with type 2 diabetes. At least 150 min of physical activity per week and dietary regulation can prevent or delay the incidence of type 2 diabetes in high-risk and prediabetic patients [7]. Patients with type 1 diabetes mellitus are sensitive to blood glucose changes during physical activity [8].

Multiple mice and rat animal studies demonstrated that exercise exerts beneficial effects on diabetes by improving insulin secretion function, islet composition, beta-cell morphology, or glucose tolerance [9]. There is also one review that discussed the potential mechanism of the physical activity’s beneficial effect on the pancreatic beta-cells [10], however, the paper did not present the appropriate exercise protocols for various beta-cell functions. In this review, we summarized appropriate exercise protocols in various beta-cell statuses, aiming to provide a reference for exercise and beta-cells-related research. 

## 2. Materials and Methods

### 2.1. Data Collection

PubMed database was used to search the literature for protocols of an exercise intervention on the human, mice, and rat pancreatic beta-cells using terms as (exercise OR training OR physical activity) AND (islet OR insulin secretory function). No additional filtrate was set to maximize coverage of all relevant studies. Originally, we searched a total of 2005 articles. After excluding the non-beta-cells related articles, 84 articles were finally collected including mice-related articles (n = 13), rat-related articles (n = 48) and human-related articles (n = 23). All these articles were published before the end of March 2022.

### 2.2. Data Analysis

According to the various species, we classified the screened articles into 3 categories: humans, rats, and mice. Then, articles were further divided into subgroups based on the type of the diseases. For example, mice-related articles were further divided into 8 subgroups, including healthy, exercise-induced hyperinsulinemia, l-monosodium l-glutamate-induced obesity, paternal or maternal obesity, high-fat-diet-induced obesity, leptin-deficient induced obesity, prediabetes, and type 1 diabetes. Table 1 shows the description of the precise exercise protocols and the ranks of the retrieved studies according to forms and duration of exercise.

## 3. Results

### 3.1. Exercise-Related Effects of Mice Pancreatic Beta-Cells 

After classifying and analyzing the thirteen selected pieces of literature, it was found that exercise interventions were mainly focused on beta-cells in six various animal models. Resistance exercise was used to explore insulin secretion of beta-cells in healthy young male mice (n = 1). Acute single high-intensity treadmill exercise was used to study the pancreatic beta-cells in exercise-induced hyperinsulinism (n = 1). Swimming or wheel running was used in paternal or maternal obesity (n = 2), respectively. Long-term low-intensity swimming was used in monosodium l-glutamate-induced obesity (n = 2). Treadmill training was used in leptin-deficient obesity mice (n = 1). Long-term moderate-intensity swimming was used in both high-fat-diet-induced obesity and prediabetes (n = 1). Two types of exercise including long-term treadmill exercise and short-term wheel running were used in type 1 diabetes mellitus (n = 4).

The classification of mice exercise protocols is shown in Table 2. In this review, exercise duration less than 6 weeks was defined as short-term exercise, and exercise duration longer than 6 weeks was defined as long-period exercise. For treadmill training, maximal oxygen uptake (VO_2max_) lower than 45% was set as low intensity, VO_2max_ between 45% and 63% was set as moderate intensity, and VO_2Max_ more than 64% was set as high intensity. For swimming, the daily training time is the standard for intensity: time lower than 30 min was low intensity, the time between 30 and 59 min was moderate intensity, and time longer than 60 min was high intensity. Wheeling-running was low intensity.

#### 3.1.1. Exercise Enhanced Beta-Cells Function in Healthy Mice 

Resistance exercise exerts a beneficial effect on glycemic control and insulin-secretion function in diabetic patients [24,25]. However, is the beneficial effect due to direct roles on beta-cells or intermediate factors between beta-cells and exercise? To elucidate this question, Bronczek et al. in 2021 conducted a 10-week resistance exercise training program on healthy male mice and confirmed that exercise not only improved glucose homeostasis due to the beneficial effect on beta-cell insulin secretion function but also stimulated the secretion of serum to defend against pro-inflammation or cytokines and chemical-induced endoplasmic reticulum stress [11].

#### 3.1.2. Exercise-Induced Hyperinsulinism (EIHI)

Exercise-induced hyperinsulinism (EIHI) stimulates insulin-secreting inappropriately after vigorous exercise or pyruvate injection and is featured as an autosomal dominant disease. After linkage analysis and sequencing of two pedigrees, Pullen et al. suggested that monocarboxylate transporter-1 (MCT1, SLC16A1) mutation may be one risk factor for EIHI. To elucidate that question, they explored the correlation between EIHI and MCT1 expression in the human beta-cells [12]. Beta-cells-specific MCT1 overexpression transgenic mice were established by doxycycline injection. After one week of adaption to treadmill running, a single treadmill run with an intensity of 80% maximum running speed at an inclination of 5° was finally attained. To evaluate the hyperinsulinism induced by the exercise, the insulin secretion under pyruvate stimulation was measured and the results showed that MCT1 overexpression in beta-cells induced by exercise replicates the EIHI [12].

#### 3.1.3. Paternal Obesity

Unhealthy maternal and paternal metabolic environments intensify the onset of obese or diabetic offspring, however, maternal and paternal exercise improves the metabolic health of offspring. After 9-weeks of low-intensity (30 min/d, 3 times/week) swimming by paternal obese mice induced by a high-fat diet, it was found that glucose tolerance, insulin sensitivity, and reduced beta-cells area in male offspring could be restored [13]. Another 4-week low-intensity voluntary wheel-running in both obese fathers and mothers significantly improved the fasting glucose level (lower 54.5%), glucose tolerance (lower 200–300 mg/dL of AUC), insulin sensitivity (lower 20–30 mg/dL of AUC), as well as beta-cell mass (increased 0.2–0.3 mg) and the beta-cell size of offspring [14].

Therefore, in obesity, 4–9 weeks of low–moderate intensity swimming, voluntary roller, or treadmill exercise can improve the function of offspring beta-cells without changing the diet.

#### 3.1.4. Monosodium L-Glutamate-Obesity

Swimming can inhibit hypothalamic obesity and restore the activity of the sympathetic adrenal axis in weaning pups. In 2009, to explore the underlying mechanism, Mathias PC et al. conducted an 8-week low-intensity (15 min/d, 3 times/week) swimming training in monosodium l-glutamate-obese mice [15]. Data demonstrated that early exercise increases sympathetic adrenal axis activity and improves insulin secretion (4.2-fold lower), and insulin sensitivity (51.5% increased), thereby improving glucose metabolism. Four years later in 2013, Mathias PC et al. further investigated the potential effects of 8-weeks of low-intensity swimming training on the insulin signaling pathways of monosodium l-glutamate-obese mice. They found that swimming increased tyrosine phosphorylation of the insulin receptor substrate-1 (IRS-1) (22% increased), activated the insulin receptors pathway, and ultimately improved glucose metabolism in obese mice [16].

Therefore, 8 weeks of low-intensity swimming exercise can improve monosodium-l-glutamic acid in monosodium L-glutamate-related obesity.

#### 3.1.5. High-Fat-Diet-Induced Obesity

Chronic obesity leads to low-grade inflammation. Although the body inhibits cellular senescence and senescence-related proteins through an anti-inflammatory response, prolonged inflammation sends signals to the rest of the body. Anti-inflammatory signals depend on the integrity of the heat shock response pathway. Exercise is a powerful inducer of the heat shock response and is expected to mitigate the negative effects of obesity. Therefore, in 2020, Homem de Bittencourt, P.I., Jr. et al. investigated the effect of moderate-intensity swimming (8-weeks) on heat shock response signaling in obesity [17] and found that 8-weeks of moderate-intensity swimming protects the pancreatic beta-cells from cytokine-induced cell death (50% decrease). 

Therefore, 8 weeks of moderate-intensity swimming inhibits islet cell death by producing the anti-inflammatory factor HSR. 

#### 3.1.6. Leptin-Deficient-Induced Obesity

Though aerobic exercise has been used in the management of obesity, it has little effect on the end stage of obesity. An 8-week aerobic exercise training program was conducted with 8-week-old leptin-deficient female mice and the results showed that exercise could decrease the ratio of the islet to the pancreas area and balance the glucose level, however, it could not attenuate the body weight gain at the later stage of obesity [18]. 

#### 3.1.7. Prediabetes

Although exercise is an effective diabetic treatment, it is not clear whether exercise has a protective effect on the beta-cells function in prediabetic KKAY mice. To solve this question, in 2013, Hua Shu et al. conducted an 8-week moderate intensity (40 min/d, 7 times/week) swimming program for prediabetic mice and found that swimming improved body weight (30% decreased), fasting glucose level (50% decreased), glucose tolerance (59% decreased), insulin sensitivity (5-fold increased), pancreatic beta-cell size (50% decreased), morphology and proliferation (0.05%) of prediabetic mice, prolonging the compensatory insulin hypersecretion period and delaying the onset of disease [19]. 

#### 3.1.8. Type 1 Diabetes Mellitus

Type 1 diabetes mellitus (T1DM) is an autoimmune disease. Exercise training enhances the beta-cell mass of T1DM patients. Studies showed that 8-weeks of high-intensity (60 min/d, 5 times/week) treadmill running reduced inflammatory factors and apoptosis (15–38%) of pancreatic beta-cells through the IL-6 signaling pathway [20], a 6-week treadmill exercise program stimulated the beta-cells proliferation, inhibited cytokine-induced beta-cells death, and, finally, improved the beta-cell mass [23].

Anti-inflammatory factors, such as bioactive compounds and physical activity, may reduce or avoid the development of autoimmune diseases such as T1DM. Chronic moderate-intensity treadmill running in T1DM mice exerted a beneficial effect on the body’s immune response by reducing the invasion of islet immune cells (50% decreased) and extension of pancreatitis [21].

Exercise can also improve blood glucose levels in type 1 diabetic mice. Results indicated that 6-weeks of voluntary wheel-running significantly improved insulin content (3-fold increased) and insulin secretion (3-fold increased) of pancreatic beta-cells in type 1 diabetic mice induced by STZ [22]. Therefore, the glycemic control benefits of exercise in T1DM mice may be due to the increasing the islet insulin content and secretion.

### 3.2. Exercise-Related Effects on Rats’ Pancreatic Beta-Cells 

After classifying and analyzing the 39 selected articles, it was found that eight animal models explored the effect of exercise on pancreatic beta-cells, as shown in Table 3. One article explored the effect of low-intensity exercise on insulin-secreting beta-cells in an aging animal model [26]. Numerous studies analyzed the effect of long-duration swimming, short or long-duration treadmill running, and resistance exercise on the function and morphology of pancreatic beta-cells (n = 14). In a Zucker diabetic fatty rats prediabetic animal model (n = 3), acute single, short, and long-term treadmill running was used to explore the effect of exercise on islet beta-cells. Streptozocin (STZ)-induced type 1 diabetes mellitus rats models (n = 2) used treadmill running and swimming to explore the effects of exercise on the morphology and function of beta-cells. Type 2 diabetes mellitus (n = 7) used three animal models including Otsuka Long-Evans Tokushima fatty rats, 90% pancreatectomized diabetic male rats, and Zucker diabetic fatty (ZDF) rats, respectively and explored the effect of rotating exercises of treadmill running, swimming, and voluntary wheel running on pancreatic beta-cells. Obesity-related studies used two animal models of FA/FA Zucker rats and monosodium l-glutamate MSG induced rats to explore the effect of short or long-term swimming on pancreatic beta-cells. Paternal effects on offspring beta-cells used treadmill running or a motorized treadmill (n = 2). A metabolic syndromes model induced by fructose water, Coca-Cola, or a high-fat diet was used for long-term treadmill running or voluntary wheel running. 

Exercise protocols for rats are shown in Table 3. The definitions of exercise duration and intensity are the same as for the mice mentioned in this review.

#### 3.2.1. Effect of Exercise on Aging-Related Beta-Cells Impairment

In 1981, Reaven et al. explored the correlation between physical activity and aging impaired beta-cell function. Reaven et al. compared the effects of 3–4-month, 6–8-month, and 12-month exercise programs on islet beta-cell morphology and the insulin-secreting function respectively. Results showed hyperinsulinemia can be induced in 4-months in rats in the non-exercise group, and islet enlargement, multi-lobulation, and fibrosis can be formed within 12-months. Exercise reduced insulin requirements (2-fold at 3–4 months, 3-fold at 10–12 months) and improved hyperinsulinemia. However, it could not correct the glucose-stimulated insulin secretion [26].

#### 3.2.2. Effect of Exercise on Healthy Rats’ Beta-Cells

A total of 15 studies investigated the effects of exercise on islet beta-cells in healthy rats, containing female rats (n = 4) and male rats (n = 11). The exercise-related rat beta-cells study was first carried out in 1983. LeBlanc et al. conducted a 10-week high-intensity (120 min/d, 6 times/week) swimming program on female Wistar rats and evaluated the insulin secretion under three different stimulations (glucose, tolbutamide, and arginine) respectively. Finally, results showed that exercise training led to a decrease in plasma insulin levels under all stimulations [63]. In 1992, Carpinelli et al. explored the effect of voluntary wheel running on K^+^ stimulated glucose oxidation and insulin secretion in islets of female Wistar rats and found that although voluntary wheel running reduced the insulin-secreting function (50% decreased), glycolysis and TCA cycling levels did not change [27]. The other two articles explored islet cell changes in female Wistar rats after short and long-term high-intensity exercise respectively. Results showed that 3-weeks of high-intensity treadmill running reduced glucokinase activity (65% decreased), islet cells proinsulin (78% decreased), mRNA levels, and insulin secretion [28]. Although 10-weeks of high-intensity treadmill running did not change the islet lipid metabolism, it greatly reduced the exposure of exogenous lipids (40% decreased) to islets. Hence, exercise can prevent islet beta-cell failure and the onset of T2D [29].

Studies on exercise-related male rats’ islets began in 1981. Results showed that compared with the non-exercise group, the exercised group had a lower insulin secretion ability or higher clear insulin ability. To figure out the potential mechanisms, in 1981, Vinten et al. conducted a 12-week high-intensity (60 min/d) swimming intervention on male SD rats and found that physical exercise could lower the insulin demand (about 15% decreased) and insulin sensitivity. The down-regulating of pressure in beta-cells might be a potential mechanism to prevent diabetes [30].

Exercise exerts beneficial effects on beta-cells, however, does an anaerobic threshold training also regulate and influence insulin secretion? To answer the question, in 2010, Mello et al. conducted an 8-week high-intensity (60 min/d) swimming training on adult male Wistar rats and found that anaerobic threshold exercise increased insulin content (3-fold increased), insulin secretion, and improved glucose metabolism [31].

To answer the above question, in 1995, Farrell et al. carried out relevant studies and found that 4-days of acute resistance exercise increased the insulin adaptation response (10 mM arginine stimulated insulin secretion) [32].

Zoppi El Al et al. explored the mechanism of an 8-week endurance exercise program on insulin secretion in male Wistar rats and found that long-term endurance exercise down-regulated the insulin secretion function (about 40% decreased) and activated the adenosine monophosphate-activated protein kinase (AMPK) signaling pathway in a frequency-dependent manner [33]. In 1992, Engdahl et al. explored the effect of endurance exercise on the insulin secretion of a single islet beta-cell and found that an 11-week endurance exercise reduced insulin secretion (approx. 43% decrease). Factors such as intracellular calcium ions, beta-cell glucose transporters, secretory grain margins, or glucose metabolites may be the key factors [35]. Endurance training increased the function and quality of pancreatic beta-cells. To investigate the effects of exercise on the growth and apoptosis of islets, male Wistar rats were treated with 8 weeks of endurance training. Results showed that endurance training activated protein kinase B (AKT) and extracellular signal-regulated kinase 1/2 (ERK1/2) pathways, reduced reactive oxygen species (ROS) production and apoptosis protein content, enhanced antioxidant capacity, and ultimately promoted the growth and survival of beta-cells [64].

In 2003, Izawa et al. explored the effects of a 9-week exercise program on the activity of glucokinase (GK), nitric oxide (NO) production, glucose transporter-2 (GLUT-2), and nitric oxide synthase (NOS) expression in the islets of healthy rats, and found that exercise training reduced GSIS partly due to the reduction of glucose utilization in the first step of glycolysis [36]. In 2012, Carvalho et al. explored the correlation between physical exercise (chronic and acute response) and islet function and found that acute exhaustion training improved insulin secretion by increasing the cholinergic activity of islets [37]. In 2013, Fujii et al. explored the effect of a 12-week exercise program on high-stimulated insulin secretion and found that chronic exercise enhanced the insulin secretion ability of non-diabetic rats but did not change the insulin content. In nondiabetic rats, chronic exercise may maintain peripheral insulin sensitivity and promote insulin secretion by potassium ions [38]. In 2014, Murguia et al. found that chronic exercise increased plasma brain-derived neurotrophic factor (BDNF) levels (about 28% increase), islet size (about 30% increase), and glucose resistance (about 27% increase) in a TRKB-dependent manner [39].

In healthy rats, 8 weeks of high-intensity swimming can improve insulin secretion function, and 12 weeks of high-intensity swimming can further reduce insulin demand (about 15%) and reduce the pressure of beta-cells. Four days of acute resistance exercise can improve the insulin secretion function under arginine stimulation, an 8-week program of resistance exercise promotes the growth and survival of beta-cells and reduced insulin secretion function by 40%, and 11 weeks of endurance exercise can reduce insulin secretion function by 43%.

#### 3.2.3. Effect of Exercise on Prediabetic Beta-Cell Impairments

Two articles both explored the relationship between treadmill running and islet beta-cells in Zucker diabetic fatty rats [40,41]. In 2005, Lund et al. found that exercise improved hyperglycemia, insulin sensitivity, and beta-cell morphology in prediabetic rats and the activation of AMPK might be a therapeutic approach [40]. In 2015, Hermansen et al. conducted a 5-week moderate-intensity intervention in prediabetic rats and conducted microRNA CHIP sequencing of islets cells, which was the first and the only paper to explore the effect of an exercise intervention on transcription levels of islets [41]. Inducible carbonyl reductase (CBR) can metabolize substrates such as aromatic ketones, resulting in detoxification or inactivation of chemically active carbonyl groups. Results showed that CBR mRNA expression was decreased in trained ZDF rats, and the decreased ketone and reactive oxygen species production might be the reason for the benefits of exercise in islet toxicity and cell death. One article estimated the effect of moderate swimming on glycemic control and non-toxic oxidative stress in pregnant mildly diabetic rats and concluded that glucose tolerance and islet morphology were positively correlated with aerobic training [42].

In prediabetes, 10 days of short-term low-intensity swimming exercise improves islet toxicity and cell death by down-regulating reactive oxygen species production; 5 weeks of moderate-intensity treadmill exercise can improve beta-cell morphology and insulin sensitivity; 8 weeks of moderate-intensity swimming can reduce the fasting blood glucose level by 50%, and increase insulin sensitivity by 5 times, and; 12 weeks of resistance exercise before meals can enhance insulin sensitivity by 25%.

#### 3.2.4. Effect of Exercise on T1DM Rats Beta-Cell Impairments

Two studies investigated the effects of exercise on the morphology and function of pancreatic beta-cells in STZ-induced type 1 diabetic rats. In 2014, Melling et al. used 10-weeks of high-intensity (60 min/d, 5 times/week) treadmill running as an intervention to investigate whether aerobic training could change islet composition in T1DM rats. However, results showed that exercise did not change the islet composition in STZ-treated T1D rats. These results indicated that the blood glucose regulated by exercise is multi-dimensional, and should be comprehensively evaluated [43]. Another article found that a 4-week low-to-moderate intensity swimming program increased insulin sensitivity, glucose transporters, insulin receptors, pancreatic beta-cells proliferation, glucose uptake, and insulin secretion in vitro [44]. 

Therefore, 4 weeks of low-moderate intensity swimming promotes insulin secretion in vitro by increasing insulin sensitivity; 6 weeks of low-intensity autonomous running wheels increases the insulin content by 3 times and exerts beneficial effects; a 6–8-week high-intensity treadmill exercise program inhibits beta-cell death by secreting anti-inflammatory factors; 10 weeks of high-intensity treadmill exercise cannot change the islet composition of T1D rats, but can promote the proliferation of islet beta-cells in T1DM.

#### 3.2.5. Effect of Exercise on T2DM Rats’ Beta-Cells 

Seven studies investigated the effect of exercise on pancreatic beta-cells in type 2 diabetic rats. Three articles used Otsuka Long-Evans Tokushima fatty (OLETF) rats as T2DM animal models and a rotating wheel [45], voluntary wheel running [46], and long-term low-intensity treadmill running [47] were used as exercise intervention programs respectively. Relevant markers showed that exercise training every 2 and 3 days has the same effect as exercise training every day for preventing non-insulin-dependent diabetes mellitus (NIDDM) and islet morphology changes in sedentary male OLETF rats. In addition, weekly exercise training in this model was also effective in reducing the incidence of diabetes, increasing pancreatic beta-cells volume and insulin content, improving beta-cell mass, and reducing connective tissue in the pancreas.

Low and moderate-intensity treadmill running was used in 90% of pancreatectomized diabetic male rats. One study showed that 3-weeks of low-intensity treadmill running upregulated insulin or insulin-like growth factor 1 (IGF-1) levels and improved beta-cell function and quality through the insulin receptor substrate 2 (IRS2) [48]. Another 8-week moderate-intensity treadmill running intervention article explored upstream of the IRS2-IGF1 pathway and found that exercise activates the islet cAMP response element-binding protein, and enhances insulin receptor substrate (IRS-2) expression, thereby enhancing insulin/insulin-like growth factor-1 signal transduction and exerts beneficial roles on beta-cells proliferation [49]. 

Zucker diabetic fatty (ZDF) rats were treated with 6-weeks or 16-weeks of high-intensity swimming [50] or 6-weeks of low-intensity voluntary wheel exercise [51] in two studies, respectively. Results demonstrated that treadmill running delayed the progression of type 2 diabetes by up-regulating the proliferation and morphology of beta-cells, short-duration low-intensity wheel running improved islet failure by ensuring insulin mRNA and insulin storage.

Therefore, every 2 and 3 days, exercise training has the same effect on islet morphology improvement as daily exercise training in OLETF diabetic rats; 3 weeks of low-intensity treadmill exercise can improve the beta-cell function in 90% of pancreatectomized diabetic rats, and; 8 weeks of moderate-intensity treadmill exercise can further exert beneficial effects by activating islet cAMP response element-binding protein; 6 weeks of low-intensity autonomous wheel exercise improves islet failure by promoting insulin storage, and; a 16-week high-intensity swimming program delays the progression of type 2 diabetes by upregulating beta-cell proliferation. 

#### 3.2.6. Effect of Exercise on Obesity Rats Beta-Cells

In total, five articles used Zucker FA/FA rats and monosodium l-glutamate-induced obesity rats (MSG), and they all used swimming as an exercise intervention. In 2002, Kibenge et al. found that a 4-week moderate-intensity swimming program reduced the body weight and food intake of Zucker FA/FA rats, however, it did not influence the fasting plasma triglyceride (TG), glucose and insulin levels [52]. Monosodium l-glutamate (MSG) obesity rats used a 10-week swimming training as an intervention. After the intervention, researchers found these conclusions: (1) a 10-week aerobic exercise program improved hyperinsulinemia without affecting the balance of insulin secretion and glucose tolerance [53]; (2) a 10-week aerobic exercise program improved hyperinsulinemia (about 40% decrease) by regulating insulin secretion stimulated by 5.6 mM and 16.7 mM glucose levels [54]; (3) a 10-week swimming program reduced pancreatic beta-cells hypertrophy (about 41% decrease), promoted glucose transporter 2 expressions, improved glucose metabolism (about 43% and 19% lower insulin secretion under 8.3 mM and 16.7 mM glucose stimulation) without changing the glycolysis flux [55]; (4) a 10-week moderate-intensity swimming program increased the adrenal catecholamine levels (about 1-fold increased), improved islet reactivity of several compounds and peripheral function of insulin (about 1-fold decreased) to avoid the onset of obesity [56].

Therefore, 10 weeks of moderate-intensity swimming can improve hyperinsulinemia (about 40%); and 8 weeks of moderate-intensity aerobic exercise can reduce the area of pancreatic islet. 

#### 3.2.7. Effect of Exercise on the Onset of Pancreatic Beta-Cells

Three articles explored the effect of exercise on pancreatic beta-cells. Fetal growth restriction contributes to the reduced beta-cell mass and the impaired glucose metabolism. A 4-week treadmill running program in adulthood improved the islet surface area (about 2-fold and 11% increase at 9 weeks and 24 weeks), restored glucose tolerance, and exerted beneficial effects on the health of male Wistar-Kyoto rats [57]. In 2016, Quiclet et al. explored the influence of submaximal exercise on offspring metabolic health and found that a 4-week high-intensity treadmill running program modified temporal offspring pancreatic function (about 22% decrease for glucose tolerance) and was negative to sedentary offspring glucose handling [58]. The adverse effect of paternal obesity on adult offspring can be attenuated through maternal voluntary training. Improvement of skeletal muscle insulin resistance and beta-cell mass and function may be the potential mechanism [59].

#### 3.2.8. Effect of Exercise on Pancreatic Beta-Cells of Metabolic Syndrome Rats 

Three studies investigated the effects of exercise on pancreatic beta-cells in metabolic syndrome rats, including fructose-water induced metabolic syndrome [58], and high-fat diet-induced metabolic syndrome [62]. Results showed that a 9-week moderate-intensity treadmill running program improved pancreatic beta-cells volume densities (about 54% increased) in fructose-water induced metabolic syndrome rats [60], 24 weeks of moderate-intensity aerobic exercise exerted beneficial effects on pancreatic morphology (about 15% and 47% increase in median islet area and beta-cell mass) in Cola-beverage induced metabolic syndrome rats [61] and reduced the risks of diabetes mellitus. A 21-week low-intensity physical activity program weakened the impairments of beta-cell function, but could not prevent the onset of metabolic syndrome. 

Therefore, 9 weeks of moderate-intensity treadmill running can increase the pancreatic volume density by 54% in fructose water-induced metabolic syndrome rats, 24 weeks of moderate-intensity treadmill exercise can increase the median-size beta-cells number, and 21 weeks of low-intensity physical activity, although it can weaken the damage of high-fat to beta-cell function, it cannot prevent the counteracting of metabolic syndrome.

### 3.3. Effect of Exercise on Human Pancreatic Beta-Cells

In total, 17 studies on exercise and human islet beta-cells were obtained, containing healthy subjects [65,66,67], athletes [68,69], elderly people [70,71], prediabetes [72,73], and diabetic patients [74,75,76,77,78,79,80,81]. Table 4 shows the specific information of relevant articles.

#### 3.3.1. Effect of Regular Exercise on Islet Beta-Cells in Healthy People 

In total, three articles explored the effects of regular exercise on islet beta-cells in healthy people. The first report was published in 1990. King et al. compared the insulin secretion under glucose, arginine, and fat stimulation between endurance-trained and untrained men and concluded that regular exercise produced adaptations within beta-cells (about 64%, 66% lower) and attenuated excessive insulin secretion [67]. In 1995, Porte et al. explored the relative contribution of glucose availability, beta-cell function, and insulin sensitivity to glucose tolerance in seven healthy young men with regular aerobic exercise habits and found that regular aerobic exercise had no significant effect on insulin secretion in healthy young men. Glucose tolerances at 12 and 84 h after exercise were mainly regulated by insulin sensitivity, independent of beta-cells and glucose availability [65]. In 1998, Galbo et al. conducted a single bicycle training session (60 min,150 W) for 7 healthy subjects and measured blood glucose metabolism-related indexes before, 2 h, and 48 h after exercise, respectively. Results demonstrated that a single exercise session did not affect the response of pancreatic beta-cells to glucose-induced thermogenesis [71].

For healthy people, resistance exercise works by promoting the adaptive response of beta-cells and inhibiting the excessive secretion of insulin.

#### 3.3.2. Effect of Regular Exercise on Islet Beta-Cells in Athletes

In total, two articles explored the effects of long-term treadmill training on islet beta-cells in athletes. In 1988, Turner et al. compared the beta-cells response, plasma insulin, and c-peptide levels in six athletes (running an average of 40 miles per week). Results showed that six sedentary subjects had a wide range of fluctuating blood glucose levels, and the improved insulin sensitivity by exercise might be at the risk of decreased beta-cells glucose sensitivity [69]. In 2001, Elahi et al. screened 42 high-level female athletes and 14 sedentary women as experimental subjects, tested their glucose metabolism-related indicators and found that physical activity can improve the beta-cell efficiency of women (about 36% increase) throughout the whole age stage and ameliorate the effect of age on the decrease of insulin sensitivity of surrounding tissues [68].

#### 3.3.3. Effect of Exercise on Islet Beta-Cells in Aging

Aging is associated with glucose intolerance, insulin resistance, hyperinsulinemia, and decreased pancreatic beta-cell function. Insulin sensitivity is a determinant of beta-cell function. To reduce the effect of insulin sensitivity on beta-cell function, Abrass et al. in 1992 compared islet beta-cell function in 14 healthy elder and younger adults with a 6-month endurance training program and they found that the elder and younger adults had similar insulin sensitivity, and the decreased glucose tolerance (about 25% decrease) in elderly adults was entirely due to beta-cell dysfunction [70]. 

#### 3.3.4. Effect of Exercise on Islet Beta-Cells in Prediabetes

Prediabetes is the key risk of type 2 diabetes. It is predicted that 30% of prediabetes might develop into diabetes within ten years [82]. Pancreatic beta-cell function plays a vital role in the progress between prediabetes and diabetes. Thus, maintaining normal beta-cell function helps to avoid the occurrence of diabetes. Malin et al. conducted a 12-week supervised exercise intervention for prediabetes and evaluated the relationship between exercise and beta-cell function. Results suggested that exercise is positively correlated with beta-cell function in an exercise dose-dependent field [72]. In addition, Bittel et al. explored the importance of the exercise time further and found that pre-meal resistance exercise enhanced insulin sensitivity (about 25% increase) in prediabetic men. In total, maybe a pre-meal and higher intensity exercise can be helpful to avoid type 2 diabetes [73]. 

#### 3.3.5. Effect of Physical Activity on Islet Beta-Cells in Type 1 Diabetes Mellitus

Most researchers posted that physical inactivity may be a risk to islet autoimmunity and stimulates the development of diabetes. However, Snell et al. confirmed that low-physical activity did not contribute to the progress of diabetes. They recruited 95 children and youths who had islet autoimmunity for the research. After 7 years of follow-up, data showed no obvious co-relationship between physical inactivity and the occurrence of diabetes [78]. 

#### 3.3.6. Effect of Exercise on Islet Beta-Cells in Type 2 Diabetes Mellitus

Seven articles explored the effect of exercise on islet beta-cells in patients with type 2 diabetes mellitus and the exercise interventions included bicycle ergometer exercises and physical activities. In 1980, Ludvigsson et al. explored the effects of a single session bicycle ergometer exercise (20 min) on the function of islet beta-cells in diabetic children and found that a single exercise can up-regulate the levels of C-peptide and proinsulin in diabetic children, but the potential mechanism was not been further analyzed [74]. Van et al., 2000 suggested that regular physical activity not only increased insulin sensitivity but also decreased insulin secretion. This down-regulation may provide an additional mechanism to delay the progression of type 2 diabetes [75]. In 2004, Galbo et al. further classified the diabetic grade into the sedentary group and the training group according to the degree of beta-cell damage and conducted an ergometer cycling high-intensity exercise for 3 months. The results showed that the effect of exercise on beta-cell function was correlated with the degree of diabetes. Exercise could improve beta-cell function in mild patients, but not in severe patients [76]. In 2005, Fujimoto et al. conducted an endurance exercise intervention on 62 Japanese Americans and found that exercise improved insulin sensitivity (about 8.3% increase) in diabetic patients but had no significant effect on the beta-cell function [77]. However, Johansen et al. in 2020 [80] and Li et al. in 2021 [79] both conducted exercise training on type 2 diabetes patients and found that adaptive exercise protocol exerts beneficial effects on beta-cell function and reducing pancreatic fat content or beta-cell inflammation might be the potential mechanism. Lyngbaek et al. in 2021 drew up a protocol that combined dietary and exercise intervention as a lifestyle intervention, which can be a useful theoretical foundation for the design of exercise protocol.

In conclusion, 3 months of high-intensity cycling exercise can improve beta-cell function in mild patients but is ineffective in severe patients; an adaptive exercise regimen reduces pancreatic fat content or beta-cell inflammation. The effects of exercise on beta cells are still controversial, and the underlying mechanism still needs to be elucidated. In the future, further human experimental studies are needed to explore the correlation between exercise and beta-cells and further explore the mechanism in animal experiments, to find efficient green exercise protocols for the prevention and treatment of diabetes.

## 4. Discussion

Researchers have been aware of the beneficial effects of exercise on beta-cells since 1919. For example, Allen et al. proposed whether the increased muscle mass and activity stimulated the production of pancreatic factors through hormones or other substances [83]. In the past 100 years, multiple studies have been carried out on the effect of exercise on the health of islet beta-cells, but the precise exercise intervention protocols and the underlying mechanisms still need to be further clarified, and there are still limitations on exercise intervention and efficacy in rats and mice.

### 4.1. Limitations of Exercise Intervention in Rats and Mice Models

Currently, rat and mice animal models have been widely used in the study of exercise intervention. However, due to the differences between the physiological system of animals and humans, especially the innate immune system, these models cannot accurately reflect the actual situation of human diseases. In addition, rats and mice can only simulate most forms of human exercise, and there is an inconsistency between physiological clocks and humans. Therefore, more molecular mechanisms are needed to verify the transformation of data from animal models to humans, and humanized rat and mice models are also the development trend of human clinical research in the future.

### 4.2. Comparison of Exercise Interventions in Rats, Mice, and Humans

The 2020 CDS guidelines indicated that adult T2DM patients should take moderate-intensity (50–70% maximum heart rate) aerobic exercise at least 150 min per week (for example, 5 times/week, 30 min/d) (brisk walking, tai chi, cycling, table tennis, badminton, and golf, etc.). High-intensity sports including fast-paced dance, aerobics, swimming, cycling, football, basketball, etc.) have a great beneficial effect on the physical health and life treatment of T2DM patients. Combined resistance training with aerobic exercise can assist in achieving greater metabolic improvement.

#### 4.2.1. Aerobic Exercise

The most similar exercise intervention for rats, mice, and humans is aerobic exercise, such as treadmill running and swimming. At the same time, the above exercise interventions are simple to operate, exercise parameters can be adjusted at any time and the exercise intensity can be set from low to high according to the different experimental purposes. However, current studies lack the recognized intensity standards and classification based on maximum heart rate, maximum oxygen uptake, and maximum speed can cause confusion among researchers. 

#### 4.2.2. Resistance Training

Ladder climbing is the only resistance training protocol in rat and mice studies, and resistance intensity is set by ladder-climbing times and weight-bearing. However, due to the differences in muscle fiber migration and muscle biological function between rats, mice, and humans, the intensity and frequency of resistance training intervention need to be accurately set.

#### 4.2.3. Combined Exercise

A combination of aerobic exercise and resistance training improved islet beta-cells function more effectively in diabetic patients. Treadmill running combined with ladder climbing is the most common combined exercise in current studies on rats and mice. However, due to the limitation of the animal physiological structure, it is impossible to complete more complex and precise exercise interventions like those conducted for humans. Therefore, the development of exercise equipment suitable for different body parts of animals will be a major research direction in the future. 

### 4.3. Comparison of Exercise Intensity for Beta-Cell Function between Various Diseases

Treadmill exercise and swimming are the common types used in diabetes and moderate-high intensity exerts beneficial effects on beta-cell function. Swimming is the most frequently used exercise type for obesity in both rat and mice models. However, the intensity varies with the pathogenesis of obesity, low intensity is used for paternal and monosodium l-glutamate-induced obesity, and a moderate-high intensity is used for high-fat diet-induced obesity. Moderate-intensity treadmill training is used for offspring in rat models. Metabolic syndrome has no standard exercise intensity and the intensity varies with the methods of the disease model. 

### 4.4. Limitations of Exercise Intervention in Diabetic Patients

Diabetic patients have complications from other diseases, therefore, before formulating an exercise intervention, the physical quality and athletic ability of patients need to be evaluated. In addition, the intensity, frequency, and duration should be adapted according to the status of the patients and maximally alleviate the disease symptoms of the patients and improve their life quality.

Authors should discuss the results and how they can be interpreted from the perspective of previous studies and the working hypotheses. The findings and their implications should be discussed in the broadest context possible. Future research directions may also be highlighted.

## 5. Conclusions

To provide more information on exercise intervention for researchers related to exercise metabolism, this review summarizes the latest studies on the effects of different exercise interventions on human, rat, and mice islet beta-cells. 

Exercise intervention reviews were summarized to improve islet beta-cell health in rats, mice, and humans (Table 1, Table 2, Table 3 and Table 4). These findings are important to investigate the molecular mechanisms underlying the benefits of exercise interventions on pancreatic beta-cells, especially for the development of new and effective exercise prescriptions. Therefore, further research in this area is urgently needed.

## Figures and Tables

**Table 1 ijerph-19-07229-t001:** Exercise intervention characteristics of included mice model studies.

Disease	Training	Duration	Frequency	Intensity	Mouse Model	Effect (Positive/Negative)	Reference
Health	Resistance exercise	10 weeks	5 day/week	50–100% of the maximum carrying	8-week-old male C57Bl/6 mice	Positive	[11]
Hyperinsulinism	Treadmill exercise	Once	-	80% max running speed, 5° incline	Mct1-Luc transgenic mice	Positive	[12]
Paternal obesity	Swimming	9 weeks	3 day/week	15–30 min/day	5-week-old male C57BL/6NHsd (Harlan), HFD-induced obesity	Positive	[13]
Paternal obesity	Running wheel	4 weeks	-	Free to exercise	6–7-week-old C57BL/6J mice, HFD- induced obesity	Positive	[14]
Monosodium L-glutamate-obesity	Swimming	8 weeks	3 day/week	15 min/day	5-day-old pup, monosodium l-glutamate MSG injected	Positive	[15]
Monosodium L-glutamate-obesity	Swimming	8 weeks	3 day/week	15 min/day	5-day-old pup, monosodium l-glutamate MSG injected	Positive	[16]
High-fat-diet- induced obesity	Swimming	8 weeks	5 day/week	8–30 min/day	21-day-old weaned B6.129SF2/J mice, HFD-induced obesity	Positive	[17]
Obesity	Treadmill exercise	8 weeks	5 day/week	60 m/day	5-week-old leptin-deficient mice	Positive	[18]
Prediabetes	Swimming	8 weeks	7 day/week	40 min/day	5-week-old male Yellow KK mice, HFD-induced prediabetes	Positive	[19]
Type 1 diabetes	Treadmill exercise	8 weeks	5 day/week	60 min/day	3-week-old wild-type and IL-6 knockout (KO) C57BL/6 mice	Positive	[20]
Type 1 diabetes	Treadmill exercise	20 weeks	5 day/week	60 min/day	5-week-old non-obese diabetic (NOD) mice	Positive	[21]
Type 1 diabetes	Running wheel	6 weeks	-	Free to exercise	8-week-old male A/J mice, STZ injected	Positive	[22]
Type 1 diabetes	Treadmill exercise	6 weeks	5 day/week	60 min/day	6-week-old male C57B6/6J mice, STZ injected	Positive	[23]

**Table 2 ijerph-19-07229-t002:** Classification of mice exercises intensity.

	Intensity	%VO_2max_
Treadmill exercise	Low intensity	<45
	Moderate intensity	46–63
	High intensity	>63
	Intensity	Time (min/day)
Swimming	Low intensity	<30
	Moderate intensity	30–59
	High intensity	≥60
Running wheel	Low intensity	

**Table 3 ijerph-19-07229-t003:** Exercise intervention characteristics of the included rats model studies.

Disease	Training	Duration	Frequency	Intensity	Mouse Model	Effect (Positive/Negative)	Reference
Aging	Running wheel	1 year	-	Free to exercise	40-day-old male Sprague-Dawley rats	Positive	[26]
Health	Swimming	10 weeks	6 day/week	120 min/day	Female Wistar rats—180 g	Positive	
Health	Rotating wheel	4 weeks	7 day/week	4.6 km/day	Female Wistar rats—180 g	Positive	[27]
Health	Treadmill exercise	3 weeks	6 day/week	90 min/day	Female Wistar rats—180 g	Positive	[28]
Health	Treadmill exercise	10 weeks	4 day/week	90 min/day	Female Wistar rats—180 g	Positive	[29]
Health	Swimming	12 weeks	5 day/week	60 min/day	Male Sprague-Dawley rats (90–110 g)	Positive	[30]
Health	Swimming	8 weeks	5 day/week	60 min/day	Male Wistar rats	Positive	[31]
Health	Resistance exercise	4 day	7 day/week	4 exercise sessions (50 reps each) with increased resistance for each session (70→120→120→190 g)	Male Sprague Dawley rats—400 g	Positive	[32]
Health	Resistance exercise	8 weeks	1, 3, 5 day/week	initially for 5 min at 5 m/min, reaching 60 min at 30 m/min in the last training week	Male Wistar rats	Positive	[33]
Health	Resistance exercise	8 weeks	1, 3, 5 day/week	initially for 5 min at 5 m/min, reaching 60 min at 30 m/min in the last training week	Male Wistar rats	Positive	[34]
Health	Treadmill exercise	11 weeks	5 day/week	60 min/day	Male Sprague Dawley rats	Positive	[35]
Health	Treadmill exercise	9 weeks	5 day/week	90 min/day	Male Sprague Dawley rats	Positive	[36]
Health	Treadmill exercise	8 weeks	5 day/week	60 min/day	Male Wistar rats	Positive	[37]
Health	Treadmill exercise	6, 9, 12 weeks	5 day/week	60 min/day	Male Wistar rats	Positive	[38]
Health	Treadmill exercise	8 weeks	3 day/week	60 min/day	Male Wistar rats	Positive	[39]
Prediabetes	Treadmill exercise	Once			5-week-old male Zucker diabetic fatty (ZDF) rats	Positive	[40]
Prediabetes	Treadmill exercise	8 weeks	5 day/week	60 min/day at 25 m/min at 5% incline			
Prediabetes	Treadmill exercise	5 weeks	6 day/week	60 min/day at 20 m/min	7-week-old male Zucker diabetic fatty (ZDF) rats	Positive	[41]
Prediabetes	Swimming	10 days	6 day/week	30 min/day	Streptozotocin induced diabetes		[42]
Type 1 diabetes	Treadmill exercise	10 weeks	5 day/week	60 min/day at 27 m/min	Streptozotocin induced diabetes	Positive	[43]
Type 1 diabetes	Swimming	4 weeks	5 day/week	15 min→20 min→40 min→45 min	Streptozotocin induced diabetes	Positive	[44]
Type 2 diabetes	Rotating wheel	24 weeks	every other day, every 3 days and every 7 days	Free to exercise	Otsuka-Long-Evans-Tokushima fatty rats	Positive	[45]
Type 2 diabetes	Running wheel	6 weeks	-	Free to exercise	Otsuka-Long-Evans-Tokushima fatty rats	Positive	[46]
Type 2 diabetes	Treadmill exercise	12 or 28 weeks	5 day/week	30 min/day at 15 m/min	Otsuka Long-Evans Tokushima Fatty rats	Positive	[47]
Type 2 diabetes	Treadmill exercise	3 weeks	5 day/week	30 min/day	90% pancreatectomized diabetic male rats	Positive	[48]
Type 2 diabetes	Treadmill exercise	8 weeks	5 day/week	30 min/day at 20 m/min at 15% incline	90% pancreatectomized diabetic male rats	Positive	[49]
Type 2 diabetes	Swimming	6 weeks	5 day/week	60 min/day	7-week-old male Zucker diabetic fatty (ZDF) rats	Positive	[50]
Type 2 diabetes	Running wheel	6 weeks	-	Free to exercise	5-week-old male Zucker diabetic fatty (ZDF) rats	Positive	[51]
Obesity	Swimming	4 weeks	5 day/week	60 min/day	fa/fa and lean Zucker rats	Positive	[52]
Obesity	Swimming	10 weeks	5 day/week	60 min/day	Monosodium l-glutamate MSG (4 mg/g body weight) was injected s.c. in the 5 days old pup’s cervical area	Positive	[53]
Obesity	Swimming	10 weeks	3 day/week	30 min/day		Positive	[54]
Obesity	Swimming	10 weeks	3 day/week	30 min/day		Positive	[55]
Obesity	Swimming	10 weeks	3 day/week	30 min/day		Positive	[56]
Fetal growth restriction	Treadmill exercise	4 weeks	5 day/week	60 min/day at 20 m/min		Positive	[57]
Offspring	Treadmill exercise	4 weeks (gestation)	5 day/week	60 min/day at 25 m/min	15-week-old nulliparous female Wistar rats	Positive	[58]
Offspring	Voluntary wheel running	8 weeks	3 day/week	-	4 to 14-week-old mice with high-fat-diet	Positive	[59]
Metabolic syndrome	Treadmill exercise	9 weeks	5 day/week	60 min/day, 5 d/week, 0.3 km/h, 50–60% VO_2max_	Fructose intake in the drinking water (D-fructose, 100 g/L), for 18 weeks	Positive	[60]
Metabolic syndrome	Treadmill exercise	6 months	5 day/week	30 min, 5 d/week, 20 m/min, 10°	Cola-drinking rats	Positive	[61]
Metabolic syndrome	Running wheel	21 weeks	-	Free to exercise	Fed with fat (SFD) for 21 wk	Positive	[62]

**Table 4 ijerph-19-07229-t004:** Exercise intervention characteristics of included human model studies.

Disease	Training	Duration	Frequency	Intensity	Sample Size, Subjects Characteristics	Effect (Positive/Negative)	Reference
Health	Aerobic exercise	3–14 weeks	4–6 day/week	30–40 min/day	7 subjects, 21–30 years men	Positive	[65]
Health	Braked bicycle	60 min	once	150 W	7 subjects, 24–26 years men	Positive	[66]
Health	Resistance training	-	5 day/week	45 min/day	8 well-trained and 9 untrained	Positive	[67]
Athletes	Swimming/running/triathletes	-	5–6 day/week	12 h/week	67 subjects (53 athletes, 14 controls), 18–69 years with body mass index (BMI), 25 kg/m^2^	Positive	[68]
Athletes	Running	-	-	40 miles/week	6 subjects, veteran athletes	Positive	[69]
Health	Running or bicycling	6 months	5 day/week	45 min/week	25 subjects, 14 older-aged 61–82 years and 11 younger-aged 24–31 years	Positive	[70]
Health	Eccentric resistance exercise	once	-	10 repetitions of leg extension (right and left legs, separately)	16 subjects, 8 younger-aged 21–28 years and 8 older-aged 59–75 years	Positive	[71]
Health	Physical activity	-	-	-	58 subjects, 37 subjects are first-degree relatives of type 2 diabetes patients and 21 subjects are healthy	Positive	[75]
Prediabetes	Aerobic exercise	12 weeks	5 day/week	60 min/day	35 obese (66.8 ± 0.8 year) adults with prediabetes	Positive	[72]
Prediabetes	Resistance training	once	once	3 sets of 10–12 repetitions	10 obese (30–65 year) men	Positive	[73]
Type 1 diabetes	Physical activity	7 years	7 day/week	-	95 children and youth	Negative	[78]
Type 2 diabetes	Bicycle ergometer exercise	once	-	20 min	31 subjects, 20 non diabetic children aged 8–16 years and 11 insulin-dependent diabetic children aged 7–16 years	Positive	[74]
Type 2 diabetes	Bicycle ergometer exercise	3 months	5 day/week	30–40 min/day	24 subjects aged 46–57 years	Positive	[76]
Type 2 diabetes	Stretching exercises/endurance exercise	24 months	3 day/week	60 min/day	62 Japanese Americans (age 56.5 +/− 1.3 years) with impaired glucose tolerance	Positive	[77]
Type 2 diabetes	Aerobic exercise	6 months	3 day/week	60 min/day	106 patients with type 2 diabetes	Positive	[79]
Type 2 diabetes	Aerobic exercise and resistance training	12 months	5–6 day/week		98 individuals with type 2 diabetes	Positive	[80]
Type 2 diabetes	Aerobic exercise	16 weeks	3–6 day/week	40–60 min/day	80 patients with type 2 diabetes	Positive	[81]

## Data Availability

Not applicable.

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
