# Peer review of "Impacts of an Exercise Intervention on the Health of Pancreatic Beta-Cells: A Review"

_ijerph, 2022, doi:10.3390/ijerph19127229_

Round 1

Reviewer 1 Report

I thank the authors for adding new references and making the paper stronger. However, I have a suggestion to make the paper stronger. I noticed that different intensities were also included but there is no conclusion on the impact of exercise intensity in the discussion section like you have concluded for aerobic and resistance training. Exercise intensity has been shown to play a significant role in improving B cell function and glucose homeostasis. It would be nice if you add a paragraph where you compile all the studies and present the results on the basis of exercise intensity.

Minor comments

1) Please check the manuscript again there are multiple times where a space between words has been omitted and the name of the author is spelled with a small character rather than starting with a capital character.

Author Response

Thank you very much for your comments concerning our manuscript entitled “Impacts of exercise intervention on the health of pancreatic be-ta-cell: a review” (Manuscript number: ijerph-1708008). These comments are very valuable and helpful for improving our paper, as well as the important guiding significance to our researches in the future. We appreciate for Editor/Reviewer’s warm work earnestly, and hope that the corrections will meet with approval. Please feel free to contact us with any questions and we are looking forward to your consideration. The main corrections in the revised manuscript and the response to reviewer’s comments are as follows.

Response to the reviewer’s comments:

Review 1 comment

I thank the authors for adding new references and making the paper stronger. However, I have a suggestion to make the paper stronger. I noticed that different intensities were also included but there is no conclusion on the impact of exercise intensity in the discussion section like you have concluded for aerobic and resistance training. Exercise intensity has been shown to play a significant role in improving B cell function and glucose homeostasis. It would be nice if you add a paragraph where you compile all the studies and present the results on the basis of exercise intensity.

Reply: According to your comment, we have added the conclusion of exercise intensity section carefully in discussion part. Please check the revised manuscript. Thank you very much.

Minor comments

1) Please check the manuscript again there are multiple times where a space between words has been omitted and the name of the author is spelled with a small character rather than starting with a capital character.

Reply: According to your comment, we have checked the manuscript carefully and deletes the space between words, revised the small character of the author. Please check the revised manuscript. Thank you very much.

Reviewer 2 Report

In this manuscript, Zhang et al. review literature relating to the role of exercise in mitigating the onset and/or progression of diabetes.  While the authors have summarized the findings of numerous papers, there are no conclusions that would seek to condense the information into possible exercise recommendations.  It is clear that exercise helps to prevent diabetes and may even be able to reverse it, in the case of type II diabetes mellitus.  What is unclear is whether there is a minimum amount of exercise that may be necessary to either prevent or reverse the disease.  Furthermore, it is unclear from this review whether the consensus in the literature is that aerobic versus anaerobic exercise may be equally beneficial or whether one mode of exercise is better than another.  With this being said, it is unclear why the authors have included methods and results sections here, because this is not a meta-analysis.  To justify this way of laying out the paper, the authors would actually need to provide some analytical data including statistical analysis.  Generally, this manuscript is also unnecessarily long and there are numerous typographical errors, which should be amended prior to any publication.  Overall, the authors would benefit be putting more effort into interpreting and summarizing the data, rather than simply making large tables, which do not particularly improve the clarity of the review. 

Major comments:

Lines 99-100:  specifically which “humoral factors” and what type of ER stress?

Line 104-106:  specify the reason that Pullen et al. suggested a role for MCT1 in EIHI

Line 206:  “Table 2” should be “Table 3”

Line 364:  The title is unclear.  What does “offset pancreatic beta-cells” mean?  The word “offset” is used repeatedly throughout this article, and it is not clear what the authors mean by it.

Line 365:  What does “offset” mean here?

Lines 438-444:  authors need to make sure that they are emphasizing that they are talking about Type I diabetes, rather than merely “diabetes”.  Of course, in the case of type I diabetes, it is unlikely that physical activity could improve the disease, since there is already baseline impairment of insulin secretion due to autoimmunity.  It is important to note, here, however, that this study does not address whether physical inactivity could worsen the type I diabetes of adults.

Lines 533-534:  “pancreatic beta-cell” should be “pancreatic beta-cell dysfunction”

Minor comments:

Line 3:  “beta-cell” should be “beta-cells”

Line 7:  “medicine, regular. . . .” should be “medicine and that regular. . . .”

Line 26:  authors should state the exact change in the numbers of people with diabetes that correspond to the given timeframe?  From what year to what year?  2006 to 2019?  It is important to be specific.

Line 33:  the authors should decide whether they want to use “beta” or “b”, and be consistent throughout

Line 53:  “on the humans” should be “ on human”

Line 55:  “setto” should be “set to”

Line 89: “tensity” should be “intensity”

Lines 95-96:  the authors should arrange the sentence in the form of a question:  for example:  “However, is the beneficial effect. . . ?”

Line 114:  “Unhealth” should be “Unhealthy”

Line 114:  “offset” should be “onset”

Line 155:  “area, balance” should be “area, and balance”

Line 185: “literature” should be “articles”

Line 218:  “induced at” should be “induced in”

Line 239:  “offset” should be “onset”

Line 278:  remove “glucose”

Line 314:  remove “by” at the end of the sentence

Line 340:  “benefic” should be “beneficial”

Line 363:  “offset” should be “onset”

Line 340:  “is shown” should be “shows”

Lines 430-431:  “maintaining a well” should be “maintaining normal”

Line 440:  “snell” should be “Snell”

Line 441:  “attribute” should be “contribute”

Line 469:  “specialized in-depth” should be “elucidated” or “understood”

Note that this is just a small sample of the numerous errors in this article.  These and the others should be fixed to improve readability.

Author Response

Thank you very much for your comments concerning our manuscript entitled “Impacts of exercise intervention on the health of pancreatic be-ta-cell: a review” (Manuscript number: ijerph-1708008). These comments are very valuable and helpful for improving our paper, as well as the important guiding significance to our researches in the future. We appreciate for Editor/Reviewer’s warm work earnestly, and hope that the corrections will meet with approval. Please feel free to contact us with any questions and we are looking forward to your consideration. The main corrections in the revised manuscript and the response to reviewer’s comments are as follows.

Review 2 comment

In this manuscript, Zhang et al. review literature relating to the role of exercise in mitigating the onset and/or progression of diabetes.  While the authors have summarized the findings of numerous papers, there are no conclusions that would seek to condense the information into possible exercise recommendations.  It is clear that exercise helps to prevent diabetes and may even be able to reverse it, in the case of type II diabetes mellitus.  What is unclear is whether there is a minimum amount of exercise that may be necessary to either prevent or reverse the disease.  Furthermore, it is unclear from this review whether the consensus in the literature is that aerobic versus anaerobic exercise may be equally beneficial or whether one mode of exercise is better than another.  With this being said, it is unclear why the authors have included methods and results sections here, because this is not a meta-analysis.  To justify this way of laying out the paper, the authors would actually need to provide some analytical data including statistical analysis.  Generally, this manuscript is also unnecessarily long and there are numerous typographical errors, which should be amended prior to any publication.  Overall, the authors would benefit be putting more effort into interpreting and summarizing the data, rather than simply making large tables, which do not particularly improve the clarity of the review. 

 Reply: Thank you very much for your comments, and these comments are very valuable and helpful for improving our paper, as well as the important guiding significance to our researches in the future. According to your comment, we have modified the conclusion. About the methods and results sections here, we were learned from those review articles (J Sport Health Sci. 2019 Sep;8(5):422-441, J Sport Health Sci. 2020 Jan;9(1):53-73). Please check the revised manuscript. Thank you very much.

Major comments:

Lines 99-100:  specifically which “humoral factors” and what type of ER stress?

Reply: Thank you very much for this valuable comment. The authors have carefully revised the sentence as “stimulates the secretion of serum to defend pro-inflammation or cytokines and chem-ical induced endoplasmic reticulum stress”. Please check the revised manuscript. Thank you very much.

Line 104-106:  specify the reason that Pullen et al. suggested a role for MCT1 in EIHI

Reply: Thank you very much for this valuable comment. The authors have carefully revised the sentence as “After linkage analyzing and sequencing of two pedigrees, Pullen et al. suggested that monocarboxylate transporter-1 (MCT1, SLC16A1) promoter mutation may be one risk factor of EIHI.  and To elucidate that question, they explored the correlation between EIHI and MCT1 expression in human beta beta-cells”.

Line 206:  “Table 2” should be “Table 3”

Reply: According to your comment, we have modified “Table 2” into “Table 3”. Please check the revised manuscript. Thank you very much.

Line 364:  The title is unclear.  What does “offset pancreatic beta-cells” mean?  The word “offset” is used repeatedly throughout this article, and it is not clear what the authors mean by it.

Reply: According to your comment, we have modified “offset” into “onset” throughout the article. Please check the revised manuscript. Thank you very much.

Line 365:  What does “offset” mean here?

Reply: According to your comment, we have modified “offset” into “onset” throughout the article. Please check the revised manuscript. Thank you very much.

Lines 438-444:  authors need to make sure that they are emphasizing that they are talking about Type I diabetes, rather than merely “diabetes”.  Of course, in the case of type I diabetes, it is unlikely that physical activity could improve the disease, since there is already baseline impairment of insulin secretion due to autoimmunity.  It is important to note, here, however, that this study does not address whether physical inactivity could worsen the type I diabetes of adults.

Reply: Thank you very much for this valuable comment. The authors have revised “diabetes” into “Type 2 diabetes”. Please check the revised manuscript. Thank you very much.

Lines 533-534:  “pancreatic beta-cell” should be “pancreatic beta-cell dysfunction”

Reply: According to your comment, we have modified “pancreatic beta-cell” into “pancreatic beta-cell dysfunctio”. Please check the revised manuscript. Thank you very much.

Minor comments:

Line 3:  “beta-cell” should be “beta-cells”

Reply: According to your comment, we have modified “beta-cell” into “beta-cells”. Please check the revised manuscript. Thank you very much.

Line 7:  “medicine, regular. . . .” should be “medicine and that regular. . . .”

Reply: According to your comment, we have modified “medicine, regular. . . .” into “medicine and that regular. . . .”. Please check the revised manuscript. Thank you very much.

Line 26:  authors should state the exact change in the numbers of people with diabetes that correspond to the given timeframe?  From what year to what year?  2006 to 2019?  It is important to be specific.

 Reply: According to your comment, we have changed “from 246 million in 2006 during 13 years” into  “from 246 million from 2006 to 2019” throughout the manuscript. Please check the revised manuscript. Thank you very much.

Line 33:  the authors should decide whether they want to use “beta” or “b”, and be consistent throughout

 Reply: According to your comment, we have modified “b” into “beta” throughout the manuscript. Please check the revised manuscript. Thank you very much.

Line 53:  “on the humans” should be “ on human”

Reply: According to your comment, we have modified “on the humans” into “on human”. Please check the revised manuscript. Thank you very much.

Line 55:  “setto” should be “set to”

Reply: According to your comment, we have modified “setto” into “set to”. Please check the revised manuscript. Thank you very much.

Line 89: “tensity” should be “intensity”

Reply: According to your comment, we have modified “tensity” into “intensity”. Please check the revised manuscript. Thank you very much.

Lines 95-96:  the authors should arrange the sentence in the form of a question:  for example:  “However, is the beneficial effect. . . ?”

Reply: According to your comment, we have modified “However, the beneficial effect. is” into “However, is the beneficial effect. . .”. Please check the revised manuscript. Thank you very much.

Line 114:  “Unhealth” should be “Unhealthy”

Reply: According to your comment, we have modified “Unhealth” into “Unhealthy”. Please check the revised manuscript. Thank you very much.

Line 114:  “offset” should be “onset”

Reply: According to your comment, we have modified “offset” into “onset”. Please check the revised manuscript. Thank you very much.

Line 155:  “area, balance” should be “area, and balance”

Reply: According to your comment, we have modified “area, balance” into “area, and balance”. Please check the revised manuscript. Thank you very much.

Line 185: “literature” should be “articles” 

Reply: According to your comment, we have modified “literature” into “articles”. Please check the revised manuscript. Thank you very much.

Line 218:  “induced at” should be “induced in”

Reply: According to your comment, we have modified “induced at “articles” into “induced in”. Please check the revised manuscript. Thank you very much.

Line 239:  “offset” should be “onset”

Reply: According to your comment, we have modified “offset” into “onset”. Please check the revised manuscript. Thank you very much.

Line 278:  remove “glucose”

Reply: According to your comment, we have deleted “glucose”. Please check the revised manuscript. Thank you very much.

Line 314:  remove “by” at the end of the sentence

Reply: According to your comment, we have deleted “by”. Please check the revised manuscript. Thank you very much.

Line 340:  “benefic” should be “beneficial”

Reply: According to your comment, we have modified “benefic” into “beneficial”. Please check the revised manuscript. Thank you very much.

Line 363:  “offset” should be “onset”

Reply: According to your comment, we have modified “offset” into “onset”. Please check the revised manuscript. Thank you very much.

Line 340:  “is shown” should be “shows”

Reply: According to your comment, we have modified “is shown” into “shows”. Please check the revised manuscript. Thank you very much.

Lines 430-431:  “maintaining a well” should be “maintaining normal” 

Reply: According to your comment, we have modified “maintaining a well” into “maintaining normal”. Please check the revised manuscript. Thank you very much.

Line 440:  “snell” should be “Snell”

Reply: According to your comment, we have modified “snell” into “Snell”. Please check the revised manuscript. Thank you very much.

Line 441:  “attribute” should be “contribute”

Reply: According to your comment, we have modified “attribute” into “contribute”. Please check the revised manuscript. Thank you very much.

Line 469:  “specialized in-depth” should be “elucidated” or “understood”

Reply: According to your comment, we have modified “specialized in-depth” into “elucidated”. Please check the revised manuscript. Thank you very much.

Note that this is just a small sample of the numerous errors in this article.  These and the others should be fixed to improve readability.

Reply: According to your comment, a native English speaker Kenny helped us edit the manuscript. Please check the revised manuscript. Thank you very much.

Round 2

Reviewer 1 Report

I thank the authors for answering my comments. But may be there was a bit of misunderstanding in my previous comments. I meant that the exercise intensity can have its own paragraph in the results section and a short description in the conclusion. Now it looks like you have written everything in the conclusion and it has become so long. Can you please move everything to the results section and add a paragraph there like you have done for other results such as shown below  

3.1 Exercise-related effects of mice pancreatic beta-cells

3.2 Exercise-induced hyperinsulinism (EIHI)

3.... Exercise and intensity

3.1.6 Leptin-deficient induced obesity

Author Response

 Reply: Thank you very much for your comments. According to your comment, we have moved the conclusions of exercise intensity into its own paragraph in the results section and gave a short description in the conclusion. Please check the revised manuscript. Thank you very much.

Reviewer 2 Report

The authors appear to have improved the manuscript.  It now provides a relatively clear reference for the beneficial role of exercise on the development of diabetes.  It should be a useful review for people in the diabetes and exercise research communities.

Author Response

 Reply: Thank you very much for your comments, and these comments are very valuable and helpful for improving our paper, as well as the important guiding significance to our researches in the future. Thanks all you have done for us. Best wishes for you.

This manuscript is a resubmission of an earlier submission. The following is a list of the peer review reports and author responses from that submission.

Round 1

Reviewer 1 Report

I congratulate the authors for organizing such a nice review regarding the effects of exercise on Beta-cell function. I have some reservations before I can recommend the paper.

1) There are many new human studies most recent as 2021 which have also shown the positive effects of exercise of beta cell function 

-https://doi.org/10.1186/s13063-021-05207-7

  • doi: 10.1152/ajpendo.00260.2013

These two are just examples of the studies which could have been included in your review. It would make your paper much stronger. I suggest you go back and do the search again and include these new papers as well.

Reviewer 2 Report

The quality of written English in this paper is very low. At least 75% of sentences contain grammatical errors. The entire paper requires detailed professional editing, by a native English speaker who also understands science,  to improve the quality of the written English. This should have been done prior to submission. In its present form, the writing is so unclear that it was difficult to understand what the authors meant.

The authors have done a good job of describing the exercise interventions in the various studies.  However, their descriptions of results are inadequate.  It is not sufficient to say that the effect of an intervention was “positive”.  It is necessary also to indicate what method was used to quantify the variable in question (e.g. beta cell function), and to indicate the results quantitatively with  numbers and units rather than just “improved”.

The search strategy (lines 56-58) was too narrow in that it required the term “islet” for articles to to be included.  Many articles may have assessed effects on insulin secretory function without including the term “islet”. I would encourage the authors to repeat and broaden the search with the assistance of a librarian with expertise in formulating searches for systematic reviews.

SPECIFIC COMMENTS

Lines 45-46:  “thus, mixed exercise with relatively high intensity and short duration is more appropriate than aerobic exercise”.  This conclusion is not justified.  Aerobic exercise is clearly beneficial to health and is completely appropriate for people with and without type 1 diabetes. It is often necessary to reduce insulin doses and/or increase carbohydrate consumption to reduce hypoglycemia risk from aerobic exercise in people with type 1 diabetes, but this does not make aerobic exercise inappropriate for them.

Lines 48-49:  What do you mean by the terms “insulin composition” and  “glucose sensitivity” in this context?

Line 85:  What was the basis for choosing six weeks as the cutpoint to differentiate “short duration” from “long duration”.  Why not 4 weeks of 8 weeks?

Line 135: You need to define what “HSP” stands for. Earlier in the same paragraph you use the abbreviation “HSR”. Are these the same thing?  I would encourage you to spell out “Heat Shock Protein” and “Heat Shock Response” rather than using abbreviations.